# A Human Microglial Cell Line Expresses γ-Aminobutyric Acid (GABA) Receptors and Responds to GABA and Muscimol by Increasing Production of IL-8

Ashley Wagner [1], Zhimin Yan [1] and Marianna Kulka [1,2,*]

1   Nanotechnology Research Centre, National Research Council Canada, Edmonton, AB T6G 2M9, Canada; zhimin.yan@nrc-cnrc.gc.ca (Z.Y.)
2   Department of Medical Microbiology and Immunology, University of Alberta, Edmonton, AB T6G 2R3, Canada
*   Correspondence: marianna.kulka@nrc-cnrc.gc.ca

**Abstract:** Gamma-aminobutyric acid (GABA) is an essential neurotransmitter and an important regulator of neuroinflammation and disease. Microglia are important immune cells in the brain that express GABA receptors (GABAR) and respond to both GABA and GABAR agonists, yet the effect of GABA on microglial inflammatory responses is unclear. We hypothesized that GABA and GABAR agonists might modify the activation of a human microglial cell line (HMC3). We further hypothesized that *Amanita muscaria* extract (AME-1), which contained GABAR agonists (GABA and muscimol), would similarly stimulate HMC3. Ligand-gated GABAR (GABA$_A$R) and G protein-coupled GABAR (GABABR) subunit expression was analyzed by qRT-PCR, metabolic activity was determined by nicotinamide adenine dinucleotide (NADH)-dependent oxidoreductase assay (XTT), reactive oxygen species (ROS) generation was analyzed by 2′,7′-dichlorodihydrofluorescein diacetate (DCFDA), and interleukin-8 (IL-8) production was analyzed by an enzyme-linked immunosorbent assay (ELISA). HMC3 expressed several neuroreceptors such as subunits of the GABA$_A$ receptor (GABA$_A$R). HMC3 constitutively produce IL-8 and ROS. Both muscimol and GABA stimulated HMC3 to produce more IL-8 but had no effect on constitutive ROS production. GABA and muscimol altered the morphology and Iba1 localization of HMC3. GABA, but not muscimol, increased HMC3 metabolic activity. Similarly, AME-1 induced HMC3 to produce more IL-8 but not ROS and altered cell morphology and Iba1 localization. GABA induction of IL-8 was blocked by bicuculline, an antagonist of GABA$_A$R. AME-1-induced production of IL-8 was not blocked by bicuculline, suggesting that AME-1's effect on HMC3 was independent of GABA$_A$R. In conclusion, these data show that GABA and GABA agonists stimulate HMC3 to increase their production of IL-8. Mixtures that contain GABA and muscimol, such as AME-1, have similar effects on HMC3 that are independent of GABA$_A$R.

**Keywords:** microglia; γ-aminobutyric acid; muscimol; human microglial cell line 3 (HMC3); *Amanita muscaria*

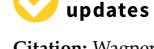



## 1. Introduction

Neuroinflammation and immune-induced changes in the brain may be responsible for migraine, depression [1] and alterations in sleeping patterns [2]. One of the most important immune cells in the brain are microglia which move throughout the brain tissue, trimming synapses [3], promoting neurogenesis, and maintaining homeostasis [4]. Gamma-aminobutyric acid (GABA) is an inhibitory neurotransmitter that regulates stress, insomnia, and fatigue and binds to ligand-gated ion channel receptors (GABA$_A$R), and G protein-coupled receptors (GABA$_B$R) [5]. GABA and the GABAergic system play an important role in inflammation, immune responses in tumorigenesis, and viral infections [6], possibly regulated by microglia. The influence of the GABAergic system on microglial function is complex and may contribute to postnatal microglial development [7], minocycline-induced

potentiation of morphine responses [8], and pronociception during diabetic neuropathic pain [9]. Mouse microglia express mRNA for both $GABA_A$ and $GABA_B$ receptors [10] but in some in vitro microglial model systems, this appears to occur only after contact with neurons [11] suggesting that complex cellular interactions in the brain mediate both GABAR expression and responses.

HMC3, also called CHME3, CHME-3, CHME-5, and C13-NJ, is a human microglial cell line that has been used by several groups as a model of human microglia [12], although its phenotype appears variable and shows some lab-to-lab inconsistency [13,14]. HMC3 cells have been reported to express CD11b and CD68 [13], but do not express the astrocyte-specific marker, glial fibrillary acidic protein (GFAP), or the neuronal neurofilament marker NF70KD [13]. Previously, we have shown that HMC3 does not express CD86, CD45, CXCR4, or CD125 but expresses toll-like receptor 3 (TLR3) and can be stimulated by poly(I:C), tumor necrosis factor (TNF), substance P, and lipopolysaccharide (LPS) to produce several cytokines [15]. Furthermore, we have previously shown that an *Amanita muscaria* extract (AME-1), a complex mixture of several compounds, is a potent stimulant of HMC3, upregulating several surface receptors and potentiating poly(I:C)-induced IL-8 production [15]. Therefore, despite its many shortcomings, HMC3 is at least an adequate model by which to measure the effect of bioactive compounds on inflammatory cytokine production. The effect of GABA and GABA agonists on HMC3 inflammatory chemokine release such as IL-8 is not known.

Compounds obtained from certain species of mushroom such as *Amanita muscaria* (fly agaric) have been proposed as effective therapies for sleep disorders, migraines, and anxiety [16,17]. These compounds include muscarine, muscimol, muscazone, and ibotenic acid [18,19] and have been used as a natural remedy in Eastern Siberia for centuries. Muscimol is a potent agonist for $GABA_A R$ which is also the target of many neurodepressants such as benzodiazepines and barbiturates [20,21]. The influence of *A. muscaria* on microglial responses and the influence of muscimol or other $GABA_A R$ agonists such as GABA on HMC3 have not been examined.

We hypothesized that HMC3 expresses GABA receptors and that GABA and muscimol could stimulate HMC3 functions, including the production of reactive oxygen species (ROS), changes in metabolic rate, and production of IL-8. Furthermore, we hypothesized that a commercially sourced *A. muscaria* extract (AME-1) which contains muscimol, an agonist of $GABA_A R$, would similarly stimulate HMC3, perhaps via the $GABA_A R$.

## 2. Materials and Methods

### 2.1. Materials

*Amanita muscaria* extract (AME-1) was provided by Psyched Wellness Inc. (Toronto, ON, Canada) and can be purchased directly from the manufacturer. AME-1 is a commercial product sold for human consumption. Gamma-aminobutyric acid (GABA) was purchased from Sigma (St. Louis, MO, USA) and muscimol from Bachem (Bubendorf, Switzerland). Phosphate-buffered saline (PBS) (Gibco, Waltham, MA, USA) was used as a vehicle for all treatments. All treatments were diluted in complete culture medium prior to their addition to cells.

### 2.2. Human Microglia Cell (HMC3) Culture

HMC3 cells were purchased from ATCC (Manasass, VA, USA). The cells were cultured in minimum essential medium (MEM) supplemented with 10% FBS (Hyclone, Logan, UT, USA), 100 U/mL penicillin, and 100 µg/mL streptomycin (Gibco) and maintained in a humidified atmosphere of 5% $CO_2$ in air at 37 °C. The cells were maintained at $2 \times 10^4$ cells/$cm^2$ and once they reached confluency, they were passaged by treating with 0.25% trypsin-EDTA (Gibco) and split at a ratio of 1:4 to 1:8 with fresh media and placed in new flasks.

*2.3. Fluorescent Microscopy and Iba1 Expression Analysis*

HMC3 cells were seeded at a density of $0.5 \times 10^5$ cells/well in a 24-well plate and left to attach for 3 h at 37 °C, 5% $CO_2$. The cells were then treated with 15 μg/mL GABA, 15 μg/mL muscimol, 500 μg/mL AME-1, or left untreated for 24 h, washed with PBS, and fixed for 15 min with 2% paraformaldehyde (PFA) in PBS. After fixation, the cells were washed with PBS, blocked for 30 min in 3% BSA/PBS, labeled with 1 μg/mL recombinant anti-Iba1 (ab178846, Abcam, Cambridge, UK) for 1 h, followed by a 45 min incubation with 10 μg/mL anti-rabbit AF 488 secondary antibody (ab150061, Abcam), and finally counterstained for 15 min with 10 mM 4,6-diamidino-2-phenylindole, dihydrochloride (DAPI, Abcam). The cells were washed with PBS in between antibody labeling and post DAPI staining. Phase contrast and fluorescent images were taken on an ECHO Revolve fluorescent microscope (ECHO, San Diego, CA, USA) at 10× magnification. Images were blind-scored by five individuals based on normal or punctate morphology and the percent punctate morphology was quantified.

For the double-blinded semi-quantitative analysis, representative images were provided to five evaluators who were blinded to the sample identity and independently scored (1) the number of cells positive for green fluorescence relative to the total number of cells (DAPI positive) and (2) the number of cells with punctate fluorescence expression relative to the total cells positive for green fluorescence, for each image. Scores were tabulated, averaged, and analyzed by a one-way ANOVA with Tukey post hoc analysis.

HMC3 was seeded at $0.4 \times 10^6$ cells per well in a 6-well plate and incubated in a humidified atmosphere of 5% $CO_2$ in air at 37 °C overnight to adhere. For single treatments, the cells were then treated with 50–5000 μg/mL AME-1, 0.15–15 μg/mL muscimol, 0.15–15 μg/mL GABA, 1 μg/mL LPS (Sigma), 10 μM bicuculline (Tocris, Minneapolis, MN, USA) or left untreated for 24 h at 37 °C, 5% $CO_2$. For co-treatments, the cells were treated with 50–5000 μg/mL AME-1 for 3 h or 10 μM bicuculline for 1 h at 37 °C, 5% $CO_2$. Following incubation, the media was aspirated, replenished with fresh media, and the cells were treated with 0.15–15 μg/mL GABA, 0.15–15 μg/mL muscimol, 50–5000 μg/mL AME-1 for 24 h. Cell-free supernatants were isolated and analyzed for human IL-8 using commercial ELISA kits (R&D Systems, Minneapolis, MN, USA).

*2.4. Metabolic Activity Analysis*

HMC3 was seeded at $0.1 \times 10^5$ cells/mL in a 96-well plate and incubated in a humidified atmosphere of 5% $CO_2$ in air at 37 °C overnight to adhere. The cells were then treated with GABA (0.15–15 μg/mL; Sigma), muscimol (0.15–15 μg/mL; Bachem), AME-1 (50–5000 μg/mL), 0.1% Triton X-100 (ThermoFisher Scientific, Waltham, MA, USA), or left untreated for 24 h. Metabolic activity was analyzed using the Cell Proliferation Kit II (XTT Assay, Roche, Basel, Switzerland) according to the manufacturer's instructions and the results were presented as percent metabolic activity relative to untreated cells.

Trypan Blue Exclusion Assay

HMC3 was seeded at $0.25 \times 10^6$ cells/well in a 6-well plate, left to adhere for 3 h, and then treated with GABA (0.15–15 μg/mL; Sigma) or muscimol (0.15–15 μg/mL; Bachem) or left untreated for 24 h. The cells were then trypsinized with 0.25% Trypsin-EDTA (Gibco) and mixed 1:1 with 0.04% Trypan blue (Gibco). Percent viable cells were counted on a hemocytometer.

*2.5. Reactive Oxygen Species (ROS) Analysis*

HMC3 was seeded at a density of $0.5 \times 10^5$ cells/well in a 96-well plate and left to adhere for 3 h. The cells were then treated with 0.15–15 μg/mL GABA, 0.15–15 μg/mL muscimol, 50–5000 μg/mL AME-1, or left untreated for 24 h. The cells were then washed in PBS, resuspended in DMEM without phenol red (Gibco) with 20 μM DCFDA and incubated at 37 °C 5% $CO_2$ for 30 min. DCF fluorescence proportional to the change in

constitutive ROS production was measured at 495/525 (ex/em) on a VarioSkan Lux plate reader (ThermoFisher).

### 2.6. qPCR Array Analysis of Genes Associated with Neurotransmitter Receptors, Neurotransmitter Synthesis, and Exocytosis

HMC3 was seeded at a density of $1 \times 10^6$ cells/mL and incubated in a humidified atmosphere of 5% $CO_2$ in air at 37 °C overnight to adhere. The cells were then treated with AME-1 (500 μg/mL) or left untreated for 3 h at 37 °C in 5% $CO_2$ and RNA was isolated using Qiagen RNeasy mini kit (Qiagen, Hilden, Germany). Complementary DNA was synthesized using 1 μg of total RNA and M-MLV reverse transcriptase (ThermoFisher). qPCR analysis was performed using 30 ng per reaction in the TaqMan Fast Neurotransmitter Array (Applied Biosystems, Waltham, MA, USA) with TaqMan Gene Expression Master Mix (Applied Biosystems) and a StepOnePlus qPCR instrument (Applied Biosystems). Results are presented as changes in CT relative to GAPDH.

### 2.7. STRING Analysis and Protein Modeling

Search Tool for the Retrieval of Interacting Genes/Proteins (STRING) analysis was performed using the online software tool which uses curated biological pathways (KEGG, Reactome, IntAct, EcoCyc, and others) to generate functional protein association networks. Analysis was performed using the full network with medium confidence (0.400) and a PPI enrichment $p$-value of less than $1 \times 10^{-16}$. The average local clustering coefficient was 0.601 and the average node degree was 13.7. Clusters were generated using kmeans clustering and an assumed number of three.

### 2.8. Metabolomic Profiling Using NMR

AME-1 was diluted 100 times with $D_2O$ containing 1.74 mM TMSP (3-(trimethylsilyl) propionate-2,2,3,3-d4) for NMR measurement. NMR experiments were performed on a Varian Direct Drive VNMRS 600 spectrometer, operating at a magnetic field strength of 14.1 T (599.49 MHz proton frequency) and equipped with an autoX dual broadband probe. One-dimensional (1D) $^1H$ NMR spectra were measured for all samples using 1D $^1H$ with water suppression sequence (NOESY 1D) at 298 K [22]. TMSP was used as an internal reference standard for spectra calibration. Metabolites were identified and quantified using Chenomx NMR Suite 8.6 Professional (ChenomX Inc., Edmonton, AB, Canada). For some metabolites, such as trehalose, GABA, and muscimol, the identification was further confirmed by spiking NMR analysis.

### 2.9. Chromatographic Conditions for HPLC-MS Analysis

HPLC-MS with electrospray ionization (ESI) was conducted on an Agilent 1260 HPLC system. For the GABA determination, the chromatographic separation was performed with an Agilent C18 column (4.6 × 250 mm, 5 μm). The mobile phase was composed of 98:2 water:methanol (EMD Millipore Corporation, Billerica, MA, USA) with 0.1% formic acid (HPLC grade, EM Science) at a flow rate of 0.1 mL/min. The volume of injection was 10 μL. For muscimol analysis, the chromatographic separation was carried out on an Agilent Zorbax $NH_2$ column (4.6 × 250 mm, 5 μm). The mobile phase was composed of 98:2 water:methanol (EMD Millipore Corporation) with 0.1% formic acid (HPLC grade, EM Science) at a flow rate of 0.1 mL/min. The volume of injection was 10 μL. Analyte detection was achieved using an Agilent Single-Quad MSD (G6135B) with an electrospray ionization (ESI) source. Identification and quantification were achieved by comparison of peak retention time, spiking LC, and area of reference standards.

### 2.10. Statistical Analysis

Experiments were conducted in triplicate and values represent the mean of n = 3 ± standard error of the mean. $p$-values were determined by one-way ANOVA with Tukey or Dunnett's

multiple comparison post hoc analysis for single treatments and two-way ANOVA with Tukey post hoc analysis for co-treatments.

## 3. Results

### 3.1. HMC3 Express Neuroreceptors

The expression of GABAR and other neuroreceptors and associated genes by HMC3 has not been extensively characterized. A qRT-PCR array analysis of over 80 genes associated with neurotransmitter receptors, neurotransmitter synthesis, and neurotransmitter exocytosis showed that HMC3 express mRNA of several families of receptors such as subunits for cholinergic receptors, GABA$_A$R, GABA$_B$R, glycine receptors, glutamate receptors, 5-HT receptors, solute carriers, syntaxins, synapsins, and neurotransmitter-associated enzymes (Figure 1A). HMC3 was particularly rich in transcripts for cholinergic receptor subunits (such as CHRNB2, CHRNB3, and CHRNB4) and enzymes (ABAT, ACHE, BSN, GAD1, and GCH1). HMC3 also expressed relatively high amounts of GABRA2, GABRA4, and GABRB1 compared to GAPDH. Notably absent from the HMC3 transcriptome (from all three RNA samples) was CHRNA2, DRD3, GABAA1, GABRA1, GABRA6, GABRG1, GABRG2, GABRG3, GABRR1, GLRA2, HTR1E, HTR2A, HTR3A, HTR3B, and SYN3. HMC3 also did not express mRNA for solute carrier subunits SLC32A1, SLC5A7, SLC6A11, SLC6A18, SLC6A2, SLC6A20, SLC6A5, SLC6A7, or SLC6A.

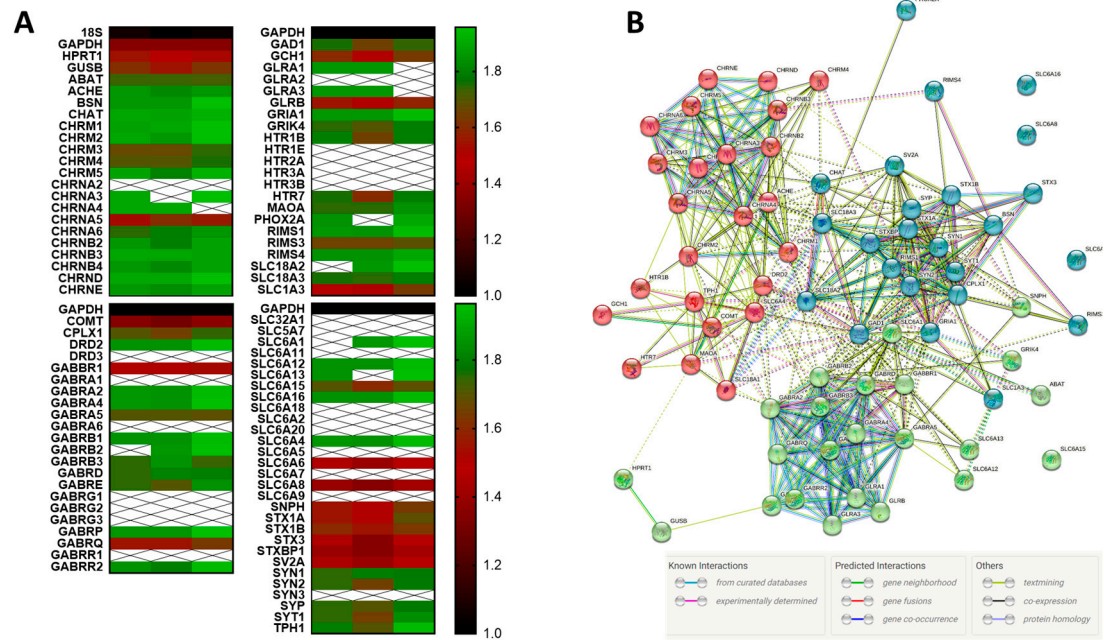

**Figure 1.** HMC3 expresses neuroreceptor and neurotransmitter-associated genes. (**A**) Three independently isolated RNA samples were analyzed for the expression of over 80 gene transcripts using qRT-PCR. (**A**) Heatmap analysis of gene expression in three RNA samples compared to the housekeeping gene *GAPDH* (black) showing high expression (green), low expression (red), or the absence of expression (white). (**B**) STRING analysis of GABAR subunit gene relationships, depicting co-occurrence and co-expression.

The search tool for the retrieval of interacting genes/proteins (or STRING analysis) allows for the prediction of gene co-occurrence and interrelationships using informational theory and existing databases, and can quantify this information to provide models of gene interactions [23]. STRING analysis of the qRT-PCR transcripts revealed three main clusters of neuroreceptor- and neurotransmitter-associated genes expressed by HMC3 that were enriched in acetylcholine receptor subunits, GABAR subunits, and syntaxins (Figure 1B). These gene clusters were associated with specific biological processes and

molecular functions such as synaptic transmission, GABA biosynthesis, GABA secretion, and GABA import (Table 1).

**Table 1.** Functional enrichments in the STRING network—top 10 biological processes and molecular functions.

| Description | Count in Network [1] | Strength [2]/False Recovery Rate [3] |
|---|---|---|
| Synaptic transmission involved in micturition | 3 of 3 | $2.44/8.19 \times 10^{-5}$ |
| GABA biosynthesis process | 3 of 3 | $2.44/8.19 \times 10^{-5}$ |
| Negative regulation of GABA secretion | 2 of 2 | $2.44/4.70 \times 10^{-3}$ |
| Behavioral response to nicotine GABA import | 4 of 7 | $2.29/6.44 \times 10^{-8}$ |
| Phospholipase C-activating G protein-coupled acetylcholine receptor signaling | 2 of 3 | $2.26/7.50 \times 10^{-3}$ |
| Adenylate cyclase-inhibiting G protein-coupled acetylcholine signaling | 5 of 8 | $2.24/1.02 \times 10^{-7}$ |
| Exocytic insertion of neurotransmitter receptor to postsynaptic membrane | 3 of 5 | $2.22/2.00 \times 10^{-4}$ |
| Regulation of GABA secretion | 4 of 7 | $2.2/5.99 \times 10^{-6}$ |
| Synaptic transmission, glycinergic | 3 of 6 | $2.14/2.90 \times 10^{-4}$ |

[1] Count in network refers to the number of genes in the network that are annotated with a particular term. [2] Strength refers to the strength of the enrichment effect calculated as log10 (observed/expected). [3] False recovery rate is an indication of the significance of the enrichment, presented as a *p*-value, calculated using the Benjamini–Hochberg procedure.

Further analysis of the three gene clusters indicated variable degrees of association between its members, with cluster 1 (acetylcholine receptor subunits) clustered strongly around *CHRNA3* (which encodes for the neuronal acetylcholine receptor subunit alpha-3 protein) and had a clustering coefficient of 0.783 (Figure 2). By comparison, cluster 3, containing the syntaxins and synaptotagmins was similar with a clustering coefficient of 0.73. The GABAR cluster had the highest co-occurrence (clustering coefficient of 0.825), and showed a high degree of association between *GABRA2*, *GABRA3*, *GABRA4*, *GABRA5*, *GABABR1*, and *GABRB3*, all encoding for α2, α3, α4, α5, β1, and β3 subunits of the GABA$_A$R.

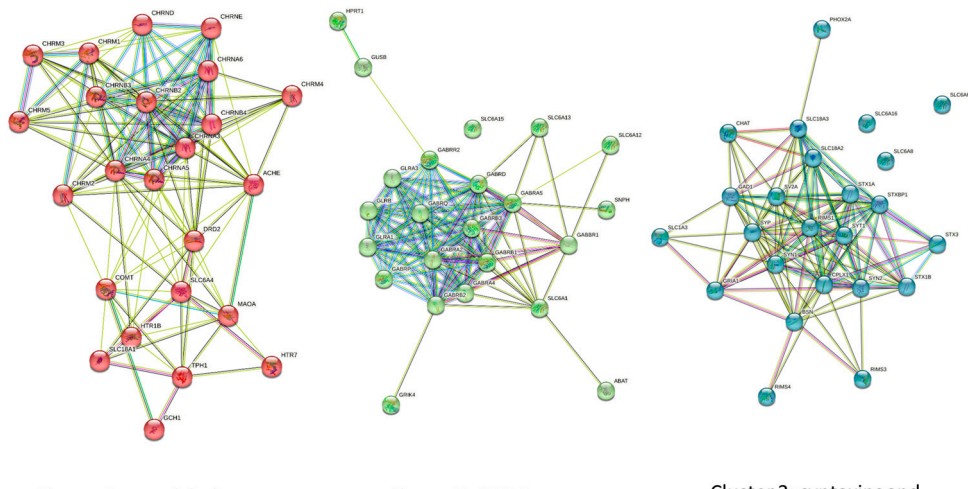

Cluster 1: acetylcholine receptors     Cluster 2: GABA receptors     Cluster 3: syntaxins and synaptotagmins

**Figure 2.** Gene clusters as determined by STRING analysis, indicating three clusters of high association. Relationships between genes is depicted by colored lines as described in Figure 1B.

The GABA$_A$R is a pentameric receptor composed primarily of the α1β2γ2 subunits but other configurations are possible, including incorporation of the ρ subunit. Our data

showed that HMC3 expresses α2, α4, α5, and β3, suggesting that HMC3 may express GABA$_A$R that contains these subunits.

### 3.2. GABA and Muscimol Cause Changes in HMC3 Morphology and Iba1 Localization

Since the STRING analysis indicated that GABA and GABA signaling were particularly important gene transcripts in HMC3, we hypothesized that HMC3 may respond to GABAergic signals and, therefore, the effect of GABA and the GABA$_A$R agonist, muscimol, on HMC3 morphology was determined (Figure 3). Phase contrast microscopy indicated that untreated HMC3 was elongated cells with some thin processes that extended away from the main cell body (Figure 3A). Some of the cells were round and appeared to have projections that radiate upward, forming a halo effect when analyzed using phase contrast microscopy. In comparison, GABA and muscimol treatment increased the number of round cells and reduced the number of cells with extensions (Figure 3B,C). The effect of muscimol was particularly interesting since some of the cells appeared to become darker and more granular.

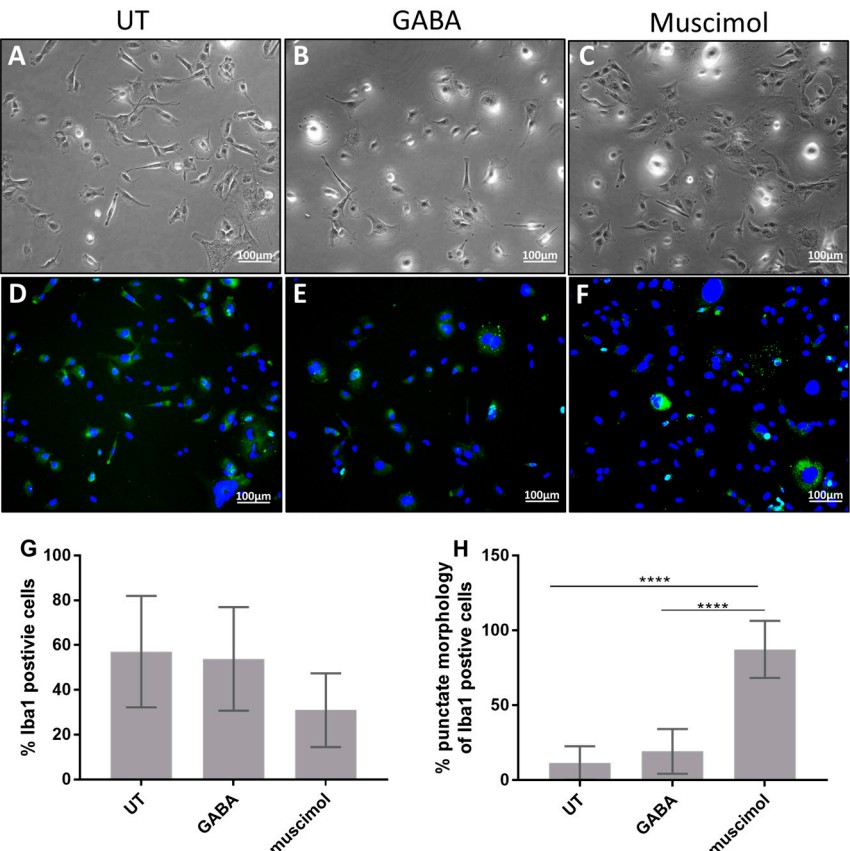

**Figure 3.** GABA and muscimol effects on HMC3 morphology and Iba1 expression. HMC3 was treated with 15 μg/mL GABA, 15 μg/mL muscimol, or left untreated for 24 h, labeled with anti-Iba1 and DAPI, and analyzed via phase contrast and fluorescence microscopy. (**A,D**) Untreated HMC3 (**B,E**) HMC3 treated with muscimol (**C,F**) and HMC3 treated with GABA. All images are representative of three separate experiments at 10× magnification. (**G,H**) Double-blinded scoring of cells positive for Iba1 expression and punctate expression within each Iba1-positive cell. Data were analyzed by one-way ANOVA with Tukey post hoc analysis and are presented as mean ± SEM of n = 3, $p \leq 0.0001$ (****).

Approximately 60% of the cells showed fluorescence associated with ionized calcium-binding adaptor molecule 1 (Iba1, or allograft inflammatory factor 1), an actin-binding protein that is a well-established marker for microglia and macrophages [24,25]. Iba1 fluorescence was diffusely dispersed throughout the cytoplasm of most cells, suggesting

the lack of specific compartmentalization of the protein in untreated cells (Figure 3D). A small portion of the untreated cells appeared to be completely devoid of Iba1 expression.

Upon treatment with GABA and muscimol, the percentage of cells that expressed Iba1 did not significantly change (Figure 3E–G). Muscimol treatment appeared to decrease the number of Iba1-positive cells, but this was not statistically significant in our analysis (Figure 3G). GABA treatment did not have a significant effect on the number of Iba1-positive cells or their localization. However, muscimol treatment measurably changed the localization of Iba1 in HMC3 since Iba1 fluorescence became more punctate, with distinct pockets of Iba1 concentrated around the nuclei (Figure 3F,H). Much of the increased Iba1 fluorescence was associated with multicellular aggregates. Some GABA or muscimol-treated cells did not show any Iba1 fluorescence (Figure 3E,F).

Since metabolic activity is an important aspect of microglial activation, the effect of GABA and muscimol on HMC3 mitochondrial activity was measured by determining NADH-dependent oxidoreduction of XTT which occurs during the pentose phosphate pathway (Figure 4A). Muscimol had no effect on metabolic activity of HMC3, and although GABA at the highest concentration (15 µg/mL) appeared to increase metabolic activity relative to the untreated control cells, this effect was not statistically significant. Trypan blue exclusion analysis further confirmed that neither GABA nor muscimol, at all of the three concentrations tested, had any effect on cell viability after 24 h treatment (Figure 4B). These data suggest that GABA and muscimol do not alter the viability or metabolic activity of HMC3.

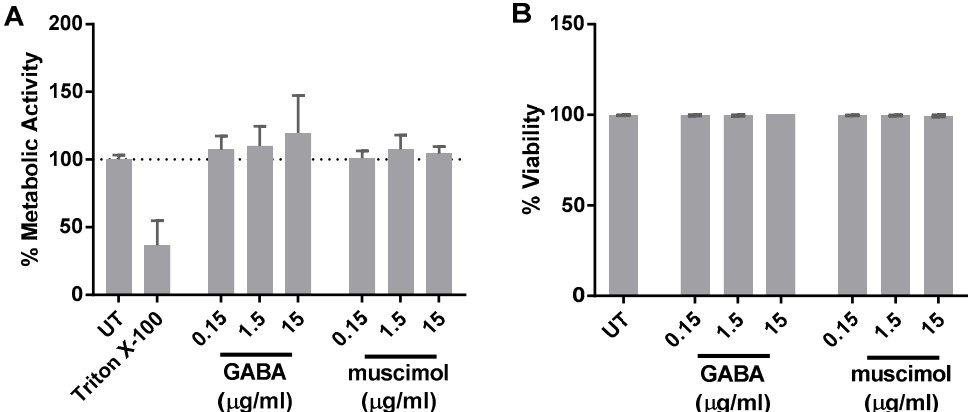

**Figure 4.** GABA and muscimol effects on metabolic activity and viability. HMC3 was treated with 0.15–15 µg/mL GABA and muscimol for 24 h and metabolic activity was measured via (**A**) XTT assay and (**B**) trypan blue exclusion assay. Untreated (UT) and triton-X-100-treated cells were used as negative and positive control, respectively. Data were analyzed by one-way ANOVA with Tukey post hoc analysis and are presented as mean ± SEM of n = 3.

### 3.3. GABA and Muscimol Have No Effect on HMC3 ROS Generation

ROS production is an important function of primary microglial cells and HMC3 cells constitutively produce significant amounts of ROS [26] which increases in response to LPS [27]. Thus, the effect of GABA and muscimol on ROS generation of HMC3 was determined. HMC3 produces a significant amount of constitutive ROS but neither GABA nor muscimol had any effect on this ROS production (Figure 5).

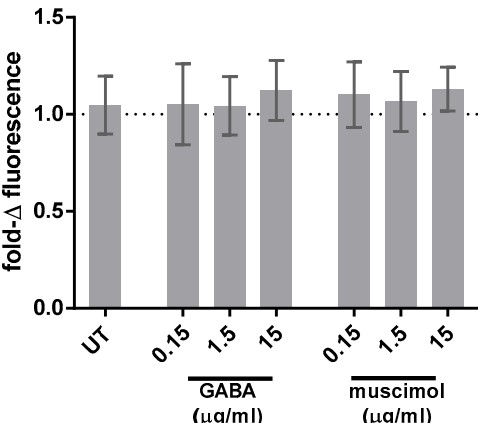

**Figure 5.** GABA and muscimol have no effect on ROS generation. HMC3 was treated with 0.15–15 µg/mL of GABA, 0.15–15 µg/mL muscimol or left untreated (UT) for 24 h and subsequently treated with 20 µM DCFDA for 30 min. The fluorescence of the deacetylated DCF compound was quantified to assess changes in constitutive ROS production and results were presented as fold change in fluorescence relative to untreated. Data were analyzed by one-way ANOVA with Tukey post hoc analysis and are presented as mean ± SEM of n = 3.

### 3.4. Muscimol and GABA Stimulate HMC3

In our previously published data, we have shown that HMC3 constitutively produces IL-8 and that several stimuli, including LPS, increase IL-8 production without significantly affecting the metabolic activity [15]. Therefore, the effect of GABA and muscimol on IL-8 production by HMC3 was determined. Confirming our previous results, HMC3 produced significant amounts of constitutive IL-8 (approximately 200 pg/mL) and both GABA and muscimol increased the production of IL-8 to approximately 300 pg/mL (Figure 6). However, increased IL-8 production did not increase with higher concentrations of GABA or muscimol.

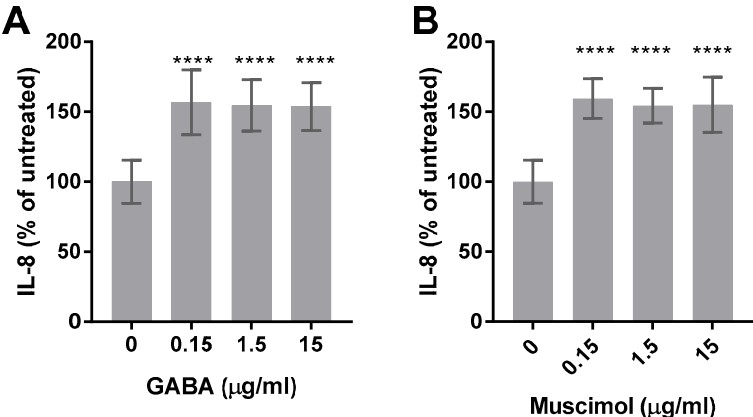

**Figure 6.** GABA and muscimol increase the production of IL-8 by HMC3. HMC3 was treated with 0.15–15 µg/mL of GABA (**A**), 0.15–15 µg/mL of muscimol (**B**), or left untreated for 24 h. Cell-free supernatants were collected and analyzed for IL-8 production via ELISA. Data were analyzed by one-way ANOVA with Tukey post hoc analysis and are presented as mean ± SEM of n = 3, $p \leq 0.0001$ (****).

### 3.5. AME-1 Contains GABA and Muscimol

Our previously published data identified some typical metabolites in AME-1 by $^1$H NMR-based metabolite profiling, suggesting the presence of GABA [15]. To determine whether GABA or muscimol were present in AME-1, we performed spiking NMR experiments and HPLC analysis (Figure 7A–D). Although the signals for GABA and muscimol were very weak, the presence of GABA and muscimol was observed. NMR profiling results

indicated that the concentrations of GABA and muscimol in AME-1 were 0.30 and 4.4 mM (Supplementary Table S1). There is more muscimol than GABA in AME-1 because the muscimol signals (4.16 and 5.84 ppm) were much larger than the GABA signal in the [1]H NMR spectra (Figure 7(Ab,Cb)).

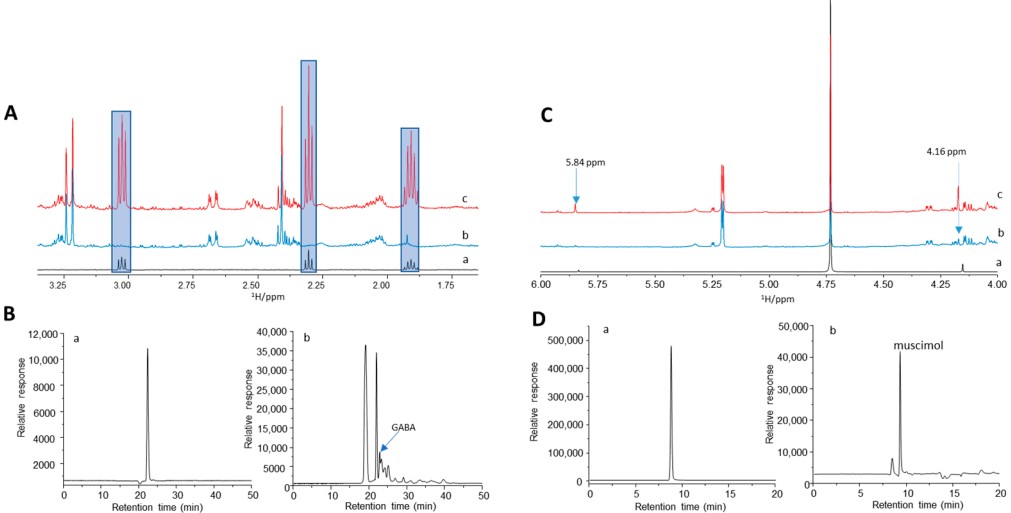

**Figure 7.** Analysis of AME-1 by NMR and HPLC. (**A**) [1]H NMR of GABA (**a**), AME-1 before (**b**) and after (**c**) spiking with GABA. The blue area highlights the signals of GABA. (**B**) HPLC-MS chromatogram of GABA: single ion monitoring mode chromatogram of GABA standard (**a**) and AME-1 50X dilute (**b**) with $m/z$ 87.1. (**C**) [1]H NMR of muscimol (**a**), AME-1 before (**b**) and after (**c**) spiking with muscimol. The signals at 5.84 and 4.16 ppm are due to muscimol. (**D**) HPLC-MS chromatogram of muscimol: single ion monitoring mode chromatogram of muscimol standard (**a**) and AME-1 50X dilute (**b**) with $m/z$ 98.1. The signals of GABA and muscimol in HPLC-MS were further confirmed by the spiked HPLC experiments (see Supplementary Information).

HPLC-MS was used to better quantify GABA and muscimol quantities in AME-1 (Figures 7B,D and S1). Straightforward HPLC analysis showed that GABA and muscimol could be separated from other metabolites and detected using a single quadrupole mass spectrometer with electrospray ionization (ESI-MS; Figures 7B,D and S1). Figure 7B shows the SIM chromatograms for GABA standard (Figure 7(Ba)) and AME-1 (Figure 7(Bb)) with $m/z$ 87.1. The peak at 22.9 min was found for GABA which was further confirmed by the spiked AME-1 sample (Figure S1c). Due to the very low level of GABA and complex metabolite composition in AME-1 as indicated by [1]H NMR, the separation of GABA in AME-1 was not very successful. Although a very good calibration curve could be constructed (Figure S1e) with a high R2 of 0.996, the accurate GABA concentration in AME-1 is uncertain. Based on the GABA integration area in AME-1, a concentration of 0.02 mM was obtained by HPLC, which is far less than 0.30 mM by [1]H NMR. This observation may also suggest that the SIM signal $m/z$ 87.1 in the AME-1 sample can react with fragments of other metabolites in AME-1. Therefore, the observed SIM $m/z$ 87.1 signal was greatly reduced.

The SIM chromatograms with $m/z$ 98.1 for muscimol standard and AME-1 are shown Figure 7D. A retention time of 9.30 min was found for muscimol. The spiked HPLC-MS (Figure S1d) clearly confirmed the peak at 9.30 min due to muscimol in AME-1. A calibration curve was constructed with a high R2 of 0.997 (Figure S1f) based on the relative response of SIM with $m/z$ 98.1. The concentration of muscimol in AME-1 was calculated to be 0.50 mM. This number is not consistent with our [1]H NMR result which indicates a concentration of 4.4 mM in AME-1. The underestimated value indicates that the SIM signal $m/z$ 98.1 in AME-1 reacts with fragments of other metabolites in AME-1.

### 3.6. AME-1 Modifies HMC3 Morphology and IBa1 Expression Localization

Human and mouse microglia change morphology when stimulated with stimulants such as LPS, muscimol, IFN-γ, IL-4, IL-10, and in some instances this induces increased adhesion and chemotaxis [28–31]. The effect of AME-1 on HMC3 morphology was analyzed by phase contrast microscopy (Figure 8). AME-1 treatment noticeably altered the morphology of the HMC3 from elongated cells with short processes radiating away from the main cell body (Figure 8A) to rounded cells with fewer processes (Figure 8B). Furthermore, the cells appeared to be more granular with a darker nucleus.

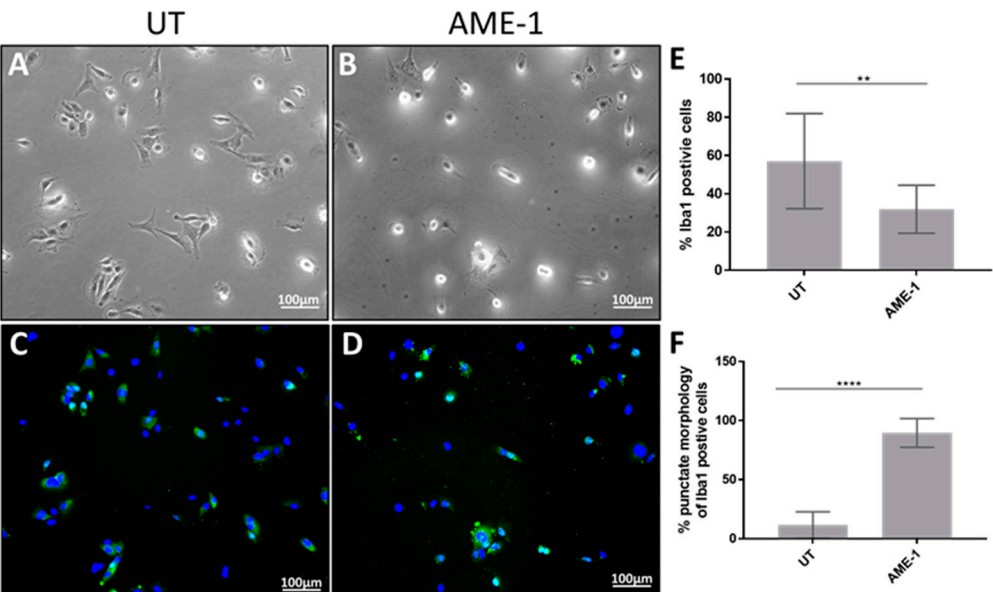

**Figure 8.** AME-1 changes HMC3 morphology. (**A**–**D**) HMC3 was treated with AME-1 (500 µg/mL) or left untreated for 24 h and labeled with anti-Iba1 and DAPI and analyzed by phase contrast and fluorescent microscopy. All images are representative of three separate experiments at 10× magnification. (**E,F**) Double-blinded scoring of cells positive for Iba1 expression and punctate expression of Iba1 relative to all Iba1-positive cells. Data were analyzed by one-way ANOVA with Tukey post hoc analysis and are presented as mean ± SEM of n = 3, $p \leq 0.05$ (**), $p \leq 0.0001$ (****). AME-1 increases IL-8 production independently of GABA$_A$R.

Immunofluorescent analysis indicated that Iba1 expression was diffuse throughout the cytoplasm in untreated cells, with the majority of cells (approximately 60%) displaying bright Iba1-associated fluorescence (Figure 8C). AME-1 treatment appeared to decrease the percentage of cell positive for Iba1 but significantly increased the number of cells with punctate Iba1 localization, particularly around the nuclei (Figure 8D). In some cells, the punctate expression was strongly associated with the DAPI-stained nucleus. In summary, the effect of AME-1 on Iba1 localization was similar to the effect of muscimol (Figure 3F,H) but more pronounced with many more cells showing the punctate localization of Iba1 around the nucleus.

Since GABA and muscimol increased IL-8 production and AME-1 contained both GABA and muscimol, the effect of AME-1 on IL-8 was determined. Similar to GABA and muscimol, AME-1 increased IL-8 production in a concentration-dependent manner (Figure 9A). Inhibition of the GABA$_A$R using bicuculline had no effect on the AME-1-mediated induction of IL-8 suggesting that AME-1 was not activating GABA$_A$R. In contrast, bicuculline blocked GABA-mediated induction of IL-8, suggesting that the GABA effect was dependent upon activation of the GABA$_A$R (Figure 9B). Since GABA and muscimol had no effect on ROS production (Figure 6), the effect of AME-1 on ROS production by HMC3 was determined (Supplementary Figure S2). AME-1 did not increase ROS production by HMC3, even at the very high concentration of 5 mg/mL (5000 µg/mL).

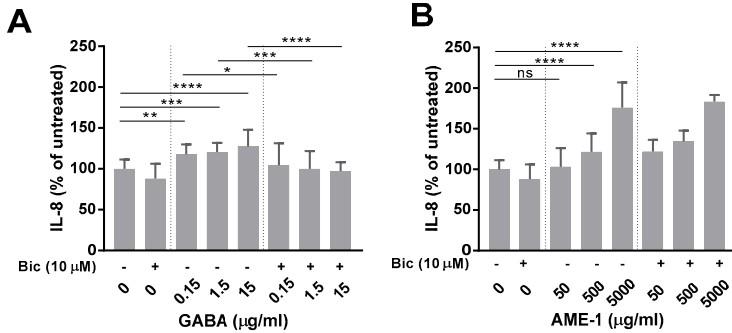

**Figure 9.** Effect of bicuculline on AME-1 production of IL-8. HMC3 was treated with 10 μM bicuculline (bic) for 1 h and subsequently treated with 50–5000 μg/mL AME-1, 0.15–15 μg/mL GABA, or left untreated for 24 h. Bicuculline, AME-1, and GABA treatments alone were used as controls. (**A**) Bicuculline does not inhibit AME-1 IL-8 production. (**B**) Bicuculline inhibits GABA-induced IL-8 production. Data were analyzed by two-way ANOVA with Tukey post hoc analysis and are presented as mean $\pm$ SEM of n = 3, $p \leq 0.05$ (*), $p \leq 0.01$ (**), $p \leq 0.001$ (***), $p \leq 0.0001$ (****), ns = not statistically significant.

### 3.7. AME-1 Modifies Neuroreceptor and Neurotransmitter Gene Expression by HMC3

To determine the effect of AME-1 on the expression of neurotransmitters by HMC3, the mRNA expression of neurotransmitter genes was evaluated by qRT-PCR (Figure 10). The analysis revealed that AME-1 selectively upregulated the expression of 5-hydroxytryptamine receptor subunits HTR1E, HTR2A, and HTR3B. Whereas untreated HMC3 did not express SLC6A18 or SLC6A20, AME-1-treated HMC3 expressed mRNA for both of these subunits. The remaining transcripts remained relatively unchanged.

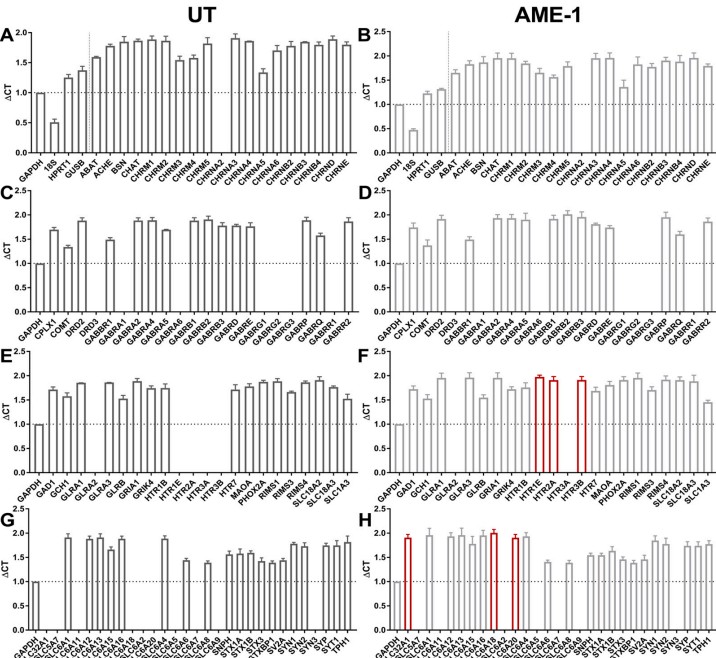

**Figure 10.** AME-1 modulates the expression of neuroreceptors and associated enzymes. HMC3 were either untrated (UT) or treated with AME-1 (500 mg/mL for 3 h) and RNA was isolated. Three independently isolated RNA samples were analyzed for the expression of over 80 gene transcripts using qRT-PCR. Data for untreated (**A,C,E,G**) and AME-1-treated samples (**B,D,F,H**) are presented as average delta Ct (critical threshold) for all three samples: expression of each gene relative to GAPDH expression for each cDNA sample (n = 3 independently generated samples from three independent treatments). Red bars indicate genes that were absent in untreated cells but expressed in AME-1-treated cells.

## 4. Discussion

In the present study, we examined HMC3 expression of mRNA for GABAR subunits and HMC3 responses to GABA and muscimol. Previously published studies had shown that although some microglial cells and cell lines respond to GABA, their expression of GABAR and other neurotransmitters has not been well characterized, and previously published data suggest that GABA can have both proinflammatory and anti-inflammatory effects on human microglia. For example, primary microglia express GABA$_A$ and GABA$_B$ receptors and release macrophage inflammatory protein-1alpha (MIP-1$\alpha$) in response to GABAergic activation [7,10] but GABA inhibits the activation of inflammatory pathways mediated by nuclear factor kappa B (NF-$\kappa$B) and p38 mitogen-activated protein (MAP) kinases [10]. Pretreatment of BV-2 microglial cells with GABAergic drugs protects them from LPS-induced injury [32], and increases in GABA signaling prevents LPS-induced microglial activation in a systemic LPS-induced model of neuroinflammation [33]. GABA activation of microglia during cold stress activates the inflammasome pathway [34], suggesting a direct link between GABAR signaling and inflammation.

We hypothesized that HMC3, like primary microglia, expresses several neurotransmitters, including GABAR, and that they would respond to GABAergic signals. Our transcriptomic analysis of over 80 genes revealed that HMC3 expresses a unique repertoire of neurotransmitter genes, including a cluster of genes that encode for GABA receptor subunits $\beta$3 and $\alpha$4. This specific GABA$_A$R configuration may make HMC3 particularly susceptible to ligands such as antidepressants, anesthetics and neuroinflammatory modulators [35–37]. A STRING network analysis revealed the GABA genes were particularly prevalent and showed a high degree of co-occurrence, emphasizing key functionalities associated with the GABAergic pathway. In addition, our analysis revealed that HMC3 expresses several other neuroreceptors and neurotransmitter-associated mRNA including acetylcholine receptor subunits, syntaxins, and synaptotagmins. Interestingly, HMC3 lacked expression of several solute carrier mRNAs and some 5-HT receptors. This is the first comprehensive characterization of neuroreceptor and neurotransmitter-associated gene expression in HMC3 although the significance of this repertoire of gene expression has yet to be determined.

Since HMC3 expressed mRNA for so many GABAR subunits, we determined the effect of GABA and the GABA$_A$R agonist, muscimol, on HMC3 morphology, Iba1 expression, and metabolism, since these were previously described as important characteristics of microglia and HMC3 [12,26,38]. GABA and muscimol caused a noticeable change in HMC3 morphology and Iba1 localization. Interestingly, GABA and muscimol increased the number of rounded cells, increased granularity, and made the cells appear darker under phase-contrast microscopy. Furthermore, muscimol caused Iba1 to colocalize around the nucleus, showing distinct punctate clusters of Iba1. Our data confirm other reports that the activation of microglia is associated with changes in morphology, alterations in Iba1 localization, and reduction in the formation of microglial processes [39,40]. Our data further show that GABA, but not muscimol, causes a slight but statistically insignificant increase in metabolic activity, although this is not associated with ROS production. Since HMC3 produces a significant amount of constitutive ROS, it may be difficult to induce further ROS production in these cells.

In order to better understand the HMC3 response to GABA, we determined whether GABA had an effect on HMC3 production of the chemokine, IL-8. Although HMC3 constitutively produced IL-8, both GABA and muscimol increased IL-8 production by almost 1.5-fold and this effect was blocked by bicuculline, a GABA$_A$R antagonist. IL-8 is not commonly associated with microglial activation and although some microglia produce IL-8 in response to certain stimuli [41–43], its significance in microglial function is unclear. Microglial cells can also produce IL-8 in response to bacterial molecules such as LPS or other cytokines such as IL-1$\beta$ and TNF [44]. In our analysis, increasing concentrations of GABA or muscimol did not significantly increase the production of IL-8 which may be due to (1) the limitation of ELISA detection and variability between experiments did not detect

small increases, (2) the maximum response occurs at or below 0.15 µg/mL, and/or (3) the GABA$_A$R response by HMC3 is an "all or nothing" response that is not concentration dependent. Additional analyses are required to examine these possibilities. Nevertheless, IL-8 is an important neutrophil chemoattractant and may be involved in neuroinflammatory diseases such as Huntington's disease [45], and several psychiatric disorders [46], but its precise role is still unknown. Since both GABA and muscimol increased IL-8 production by HMC3, this suggests that GABAergic activation amplifies already activated pathways in HMC3. Whether this is a unique feature of the HMC3 cell line, or whether this occurs in primary human microglia remains to be determined.

Lastly, we examined the effect of AME-1 on HMC3. AME-1 is a complex mixture that contains several compounds [15] and has been marketed as a supplement to promote relaxation and sleep. Our NMR and HPLC analysis indicated that AME-1 contained both GABA and muscimol as measured by NMR and HPLC. The effect of *A. muscaria* on neuroinflammatory cells, such as microglia, is unknown but since AME-1 contained GABA and muscimol we hypothesized that AME-1 would have similar effects on HMC3. Our data showed that HMC3 expresses a unique repertoire of neurotransmitters, including specific subunits of the pentameric GABA$_A$R. Like muscimol and GABA, AME-1 induced morphological changes, altered Iba1 localization, and increased HMC3 production of IL-8, but AME-1 had no effect on ROS production or metabolic rate. Furthermore, AME-1's effect on IL-8 production was not blocked by bicuculline. This data suggests that although AME-1 contains small amounts of muscimol and GABA, other components within the AME-1 extract cause increased IL-8 production and altered Iba1 localization within HMC3. Alternatively, AME-1's effect on HMC3 may be dependent upon several components acting in synergy.

In conclusion, this data shows that HMC3 expresses a unique repertoire of neuroreceptors and neurotransmitter genes, consisting of unique subunits of the GABAR. HMC3 displays a constitutively activated state—producing significant amounts of ROS and IL-8 in the absence of stimulation. When stimulated with GABA and muscimol, HMC3 cells increase the production of IL-8 but not ROS and alter their morphology and Iba1 expression, consistent with an activated state. Increased production of IL-8 in response to GABA was dependent upon GABA$_A$R. AME-1, an extract that contains both GABA and muscimol, similarly activates HMC3 to produce IL-8 but independently of GABA$_A$R. This is the first comprehensive examination of GABA and muscimol effects on HMC3.

**Supplementary Materials:** The following supporting information can be downloaded at: https://www.mdpi.com/article/10.3390/neuroglia4030012/s1, Figure S1: ESI-MS positive ion scanning spectra for GABA (a) and muscimol (b); Single ion monitoring mode chromatograms of AME-1 spiked with GABA (c) and muscimol (d); calibration curve of GABA (e) and muscimol (f) (n = 3); Figure S2: Effect of AME-1 on ROS generation. HMC3 cells were treated with AME-1 and indicated concentrations for 24 h and ROS generation was measured as indicated in Materials and Methods. Data were analyzed by one-way ANOVA with tukey post-hoc analysis and are presented as mean ± SEM of n = 3; Table S1: List of genes expressed by HMC3 as determined by the qRT-PCR array.

**Author Contributions:** Conceptualization, methodology, data generation, original draft preparation, review, and editing: A.W., Z.Y. and M.K. All authors have read and agreed to the published version of the manuscript.

**Funding:** This research was funded by the National Research Council Canada and Psyched Wellness Inc.

**Institutional Review Board Statement:** Not applicable.

**Informed Consent Statement:** Not applicable.

**Data Availability Statement:** All data are provided herein.

**Acknowledgments:** We thank Valerie Sim for providing the Iba1 antibody.

**Conflicts of Interest:** The authors declare no conflict of interest.

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
