# Peer review of "A Human Microglial Cell Line Expresses γ-Aminobutyric Acid (GABA) Receptors and Responds to GABA and Muscimol by Increasing Production of IL-8"

_2571-6980, doi:10.3390/neuroglia4030012_

Round 1

Reviewer 1 Report

In this paper, authors have evaluated the modulation of microglial cell functions, monitored as IL-8 release, provided by GABA, muscimol and the mushroom Amanita muscaria extract called AME-1 in a human microglia cell line (named HMC3 cells).

Results show that HMC3 cells express several subunits of ionotropic GABA-A and G protein-coupled GABA-B receptors. Additionally, treatment with either GABA, muscimol or AME-1 (which contains small amounts of both molecules) modulates the expression of various “neuroreceptors”, which are related to GABA and acetylcholine transmission, induced an increased release of IL-8 and morphological changes in cells.

I am afraid that these data are quite preliminary and rather confused; it is therefore very difficult to understand their novelty and scientific significance, if any. Please, see a list of specific comments below.

1. My first concern regards the choice of the in vitro model of human microglia cells, which have generated quite different data among various laboratories as admitted by authors themselves. Apart from additional criticisms below, overall the general relevance of the results obtained is very limited. For example, at least another cell line should have been analyzed.

2. The experimental design is rather confused as well. For example, no clear explanation of the choice for the array of “neuroreceptor” (which is a too generic definition) has been provided in the Methods section, and results are difficult to follow for readers who are not familial with all these genes. Additionally, although likely interesting for someone, the very long description of the analysis of AME-1 components is out of the scope of the journal (and the concentrations of muscimol and GABA are quite low, anyway).

3. Another section which is difficult to put in a more general frame is the analysis of the 3-D structure of GABA-A receptor composed by alpha5/beta3 subunits. What is the scope of this evaluation? Was this structure unknown? How can authors exclude that other subunits expressed by HMC3 cells (i.e., alpha2-4 and beta-1, as demonstrated by the authors) are included in the functional receptors? Finally, in the Discussion section (line 480) authors mention alpha4 and beta3 subunits as important cluster of gene expression; is alpha4 a mistake?

4. An observation that deserved additional in-depth analysis is the morphological changes observed by exposing HMC3 cells to either agonists or AME-1. First of all, data must be quantified to be relevant (i.e., by counting the number of cells or by evaluating their morphology by ImageJ software). Does morphology represent activation of microglia or rather cell death? Several nuclei appear shrunken and cells seem detached from the substrate, and changes in Iba1 localization seem to point toward the induction of apoptotic cell death. This must be verified in details.

5. I am sorry but I find it extremely difficult to believe that the effect of 15 ug/ml GABA on the metabolic activity of cells (Figure 5) is statistically different from UT cells. If the statistics instead refer to Triton X-100-treated samples, it must be taken into consideration the huge (and I bet statistically significant) reduction in the metabolic activity induced by the detergent alone.

6. In the Reference list, one “in press” paper from the same group is included which deals with the evaluation of the effects of AME-1 on human microglia. What are the differences with the present manuscript?

Author Response

Thank you for your careful review of our manuscript. Our responses are indicated in blue.

  1. My first concern regards the choice of the in vitro model of human microglia cells, which have generated quite different data among various laboratories as admitted by authors themselves. Apart from additional criticisms below, overall the general relevance of the results obtained is very limited. For example, at least another cell line should have been analyzed.  

    We agree that the HMC3 cell line is an imperfect model of human brain microglia. As such, we have attempted to honestly discuss both its value and its limitations in extrapolating observations to human primary microglia. We have attempted to secure another human microglial cell line to recapitulate some of the observations that we have already made using HMC3. Unfortunately, we have been unsuccessful at obtaining such a cell line within the time frame allotted for responding to the reviewers’ comments. We also do not currently have access to human primary microglia. However, we have clarified throughout the manuscript that we are using a microglial cell line, and that despite its shortcomings, it still provides some information that can be used for the design of future work with primary cell cultures.

  2. The experimental design is rather confused as well. For example, no clear explanation of the choice for the array of “neuroreceptor” (which is a too generic definition) has been provided in the Methods section, and results are difficult to follow for readers who are not familial with all these genes. Additionally, although likely interesting for someone, the very long description of the analysis of AME-1 components is out of the scope of the journal (and the concentrations of muscimol and GABA are quite low, anyway). 

    In our analysis, we used a qPCR TaqMan™ Array, Human Neurotransmitter, Fast 96-well (Catalog number: 4418739) which targets genes associated with several common groups of neurotransmitters and neurotransmitter receptors: 1) acetylcholines (CHRM and CHRN); 2) dopamine receptors; 3) GABA receptors; 4) glycine receptors; 5) glutamate receptors and 6) 5-HT (serotonin) receptors). Genes involved in neurotransmitter synthesis, transport, and degradation are represented in the panel as well. We have added this detail into the Methods section. A more comprehensive description of the genes and their role in possible microglial functions has been added to the discussion.

    The long description of the NMR data has been considerably shortened to demonstrate that the HPLC analysis of GABA and muscimol is more accurate than NMR metabolic analysis. This is important information for those in the field that may use NMR metabolomics to quantify GABA or muscimol in future samples.

  3. Another section which is difficult to put in a more general frame is the analysis of the 3-D structure of GABA-A receptor composed by alpha5/beta3 subunits. What is the scope of this evaluation? Was this structure unknown? How can authors exclude that other subunits expressed by HMC3 cells (i.e., alpha2-4 and beta-1, as demonstrated by the authors) are included in the functional receptors? Finally, in the Discussion section (line 480) authors mention alpha4 and beta3 subunits as important cluster of gene expression; is alpha4 a mistake? 

    We agree with the reviewer that this data was superfluous and unnecessary to support our conclusions. The 3D structural analysis of the GABAAR has been removed.

  4. An observation that deserved additional in-depth analysis is the morphological changes observed by exposing HMC3 cells to either agonists or AME-1. First of all, data must be quantified to be relevant (i.e., by counting the number of cells or by evaluating their morphology by ImageJ software). Does morphology represent activation of microglia or rather cell death? Several nuclei appear shrunken and cells seem detached from the substrate, and changes in Iba1 localization seem to point toward the induction of apoptotic cell death. This must be verified in details. 

    We agree with the reviewer that the Iba1 expression changes are interesting. We have performed blinded scoring of the positive cells in all of the images in our database and this data has now been presented as graphs (with statistical analysis) in Fig. 3. Interestingly, although there is a trend for a decreased Iba1 expression with muscimol treatment (although not statistically significant), there was a measurable increase in the punctate expression of Iba1 when the cells were treated with muscimol (Fig. 4H). This suggests that muscimol treatment changes the localization of Iba1 within HMC3.

    The reviewer’s comment that Iba1 may indicate apoptotic death is interesting. We have performed an additional trypan blue viability analysis which indicates that neither GABA nor muscimol have any measurable effect on HMC3 viability after a 24 hr treatment. This new data has been added to Fig. 4B. Although this isn’t a direct measure of apoptosis specifically, this data indicates that by 24 hr the cells appear viable and healthy. However, it is possible that apoptosis is occurring after 24 hr and this will be a focus of future studies outside the scope of this manuscript.

  5. I am sorry but I find it extremely difficult to believe that the effect of 15 ug/ml GABA on the metabolic activity of cells (Figure 5) is statistically different from UT cells. If the statistics instead refer to Triton X-100-treated samples, it must be taken into consideration the huge (and I bet statistically significant) reduction in the metabolic activity induced by the detergent alone.   We thank the reviewer for this comment. This was an error and the asterisk has been removed. 
  6. In the Reference list, one “in press” paper from the same group is included which deals with the evaluation of the effects of AME-1 on human microglia. What are the differences with the present manuscript?    

    In the previous manuscript, the effect of AME-1 on HMC3 activation by double-stranded RNA (dsRNA) was evaluated. The full reference has now been updated. The main observation in that manuscript was that AME-1 potentiated HMC3 responses to poly(I:C) and that this could be due to AME-1’s modulation of TLR3, RIG-I and MDA5 expression (all receptors for dsRNA). That study did not show any data pertaining to GABA, muscimol or the effect of bicuculline. The full manuscript can be found here: https://www.frontiersin.org/articles/10.3389/fphar.2023.1102465/full

Reviewer 2 Report

 First of all, I want to congratulate the authors who have carried out this experiment. 
As far as, I am concerned, it is a very interesting article, which elucidates more about microglia in the role of GABA in the neuroinflammatory process. In my view, the experimental design and the techniques are accurate and very well performed.
T
he results of the experiments are well explained and contribute to expand the knowledge of microglia cells.
Next, I am will make comments to give a constructive criticism:
- Regarding the writing: the line 310 has an extra space. - From my point of view, the discussion about changes in HMC3 morphology and Iba1 localization should be longer in connection with the release of IL-8. - On the other hand, it would be very interesting to assess how AME-1 affects the release of other interleukins with the aim of using AME-1 as a possible therapeutic target in the future.   Finally, I want to congratulate the authors again for the quality of the experiment.

Author Response

We thank the reviewer for the kind and supportive comments and the constructive feedback. We have responded to specific comments in blue below.

  1. Regarding the writing: the line 310 has an extra space.  This has been correct.
  2. From my point of view, the discussion about changes in HMC3 morphology and Iba1 localization should be longer in connection with the release of IL-8. We have added considerably more description of the Iba1 data in the results and discussion sections. In addition, we have now added a "blinded" analysis of the images in an effort to add some semi-quantitative analysis of our images. Our semi-quantitative scoring confirms that muscimol causes signficant and recognizable changes in HMC3 morphology.

  3. On the other hand, it would be very interesting to assess how AME-1 affects the release of other interleukins with the aim of using AME-1 as a possible therapeutic target in the future.  We agree that a more comprehensive analysis of AME-1's effect on the release of other cytokines by HMC3 would be interesting. We were unable to generate this data within the time alloted to respond to these reviews. However, we plan these experiments in the future -- possibly with primary microglia which would better reflect in vivo responses to AME-1.

  4. Finally, I want to congratulate the authors again for the quality of the experiment. Thank you for this supportive comment.

Reviewer 3 Report

In this manuscript, the authors look at the effects of GABA and the GABA receptor agonist muscimol as well as a mushroom extract (AME-1) that has been reported to contain both GABA and muscimol on a select number of parameters in the human HMC3 microglial cell line. On the plus side, the authors analyze the mRNA expression of a variety of different neuroreceptors and associated genes in the HMC3 cells which could be useful to others who study these cells. However, their readouts for the effects of GABA, muscimol and AME-1 on the HMC3 cells are very limited and they only look at one marker of inflammation, IL8, although they imply in the Introduction that a major goal of this study was to look at the effects of GABA and GABA agonists on the release of pro-inflammatory mediators by the HMC3 cells. A more comprehensive analysis or the effects of GABA, muscimol and AME-1 on pro-inflammatory mediator release would significantly strengthen the manuscript, especially if it was coupled with treatment with the GABA receptor antagonist. In addition, the assay that they use to measure metabolic activity is also a measure of cell proliferation and since they treat the HMC3 cells for 24 hr with either GABA or muscimol, they cannot say for sure that the increase in formazan production that they see is due to increases in metabolic activity or an increase in cell proliferation. Finally, the manuscript has a number of typographical errors that need to be corrected. Furthermore, Fig. 11 is missing so it can’t be evaluated. Additional, specific concerns are listed below.

1. Figure 4 and Figure 9: The appearance of the muscimol-treated cells appears quite different from that of the GABA-treated cells and both appear very different from the AME-1-treated cells. However, these differences are glossed over in the text. Maybe this is due to the selection of pictures but the text should reflect what is shown in the figures.

2. Figure 7: The authors need to address why they see no dose response for IL-8 production with either GABA or muscimol. It seems as if lower doses of both compounds should be tested to ensure that the increase in IL-8 is not an artifact. Does untreated mean the vehicle alone? If not, that needs to be tested as some vehicles can increase cytokine production.

3. Figure 10: The authors need to discuss why the baseline levels of IL-8 are so much higher in these studies than in the studies shown in Figure 7.

4. The authors should compare the effects of AME-1, GABA and muscimol on neurotransmitter gene expression. Only looking at the effect of AME-1 does not allow any comparison with known GABA receptor agonists and so is not particularly informative.

The English is fine but there are a number of typographical errors that should be corrected.

Author Response

We thank the reviewer for the helpful comments and suggetions for improvements to our manuscript. We have made significant changes to our manuscript and added additional data. Our specific responses are in blue below.

1. A more comprehensive analysis or the effects of GABA, muscimol and AME-1 on pro-inflammatory mediator release would significantly strengthen the manuscript, especially if it was coupled with treatment with the GABA receptor antagonist.  

We agree that it would be helpful to catalogue HMC3 production of many different mediators in response to GABA and muscimol. However, due to space and time limitations, we have focused on IL-8 since it is an important chemokine produced by macrophage-like cells to promote phagocytosis and recruit neutrophils to sites of inflammation. IL-8 may be involved in chronic inflammatory changes in neurodegenerative and neuropsychological alterations in the brain and has been implicated in several psychiatric disorders (Tsai, Progress in Neuro-Psychopharmacology and Biological Psychiatry, 2021).

2. In addition, the assay that they use to measure metabolic activity is also a measure of cell proliferation and since they treat the HMC3 cells for 24 hr with either GABA or muscimol, they cannot say for sure that the increase in formazan production that they see is due to increases in metabolic activity or an increase in cell proliferation.     

It is correct that the XTT assay, much like the MTT assay, is often used as an indirect measure of cell proliferation because increased metabolic activity is sometimes conflated with an increase in cell number. Therefore, we have performed a more basic analysis of cell viability and cell number using trypan blue and hemocytometer counting and we have observed no significant change in either viability or cell number after GABA and muscimol treatment.

3. Finally, the manuscript has a number of typographical errors that need to be corrected. Furthermore, Fig. 11 is missing so it can’t be evaluated. Additional, specific concerns are listed below. 

a.  Figure 4 and Figure 9: The appearance of the muscimol-treated cells appears quite different from that of the GABA-treated cells and both appear very different from the AME-1-treated cells. However, these differences are glossed over in the text. Maybe this is due to the selection of pictures but the text should reflect what is shown in the figures.  We have added considerably more discussion of these differences and some of the more specific features of the cells that appear to be changed upon treatment with GABA and muscimol.

b. Figure 7: The authors need to address why they see no dose response for IL-8 production with either GABA or muscimol. It seems as if lower doses of both compounds should be tested to ensure that the increase in IL-8 is not an artifact. Does untreated mean the vehicle alone? If not, that needs to be tested as some vehicles can increase cytokine production.      

The vehicle in this experiment was media and thus had no effect on background production of IL-8. We agree with the reviewer that it is curious that increasing concentrations of GABA and muscimol do not increase IL-8 production. We have tested lower concentrations of GABA and muscimol (below 0.15 ug/ml) but these had no effect on IL-8 production and therefore we chose to only show the concentrations that produced a response. The lack of a dose-response could be due to (1) limitation of ELISA detection and variability between experiments did not detect small increases, (2) the maximum response occurs at or below 0.15 ug/ml, and/or (3) the GABAAR response by HMC3 is an “all or nothing” response that is not concentration dependent. Additional analyses are required to examine these possibilities – perhaps using more sensitive measures of IL-8 such as chemiluminescent and electrochemical sensing assays. We have added these statements to the discussion.

c.  Figure 10: The authors need to discuss why the baseline levels of IL-8 are so much higher in these studies than in the studies shown in Figure 7.   

We have noticed that as the HMC3 age, they produce higher amounts of constitutive IL-8. As a result, using different passage numbers shows some variability in IL-8. To compensate for this variability in background release, we have standardized our data to the untreated control and expressed our data a “% of untreated” control.

d. The authors should compare the effects of AME-1, GABA and muscimol on neurotransmitter gene expression. Only looking at the effect of AME-1 does not allow any comparison with known GABA receptor agonists and so is not particularly informative.  

The reviewer is correct that comparison of the effect of GABA, muscimol and AME-1 on all of the 80 target genes mentioned in this study would be interesting. However, this would require a great deal of additional experiments not possible within the time frame allowed for the review response. The effect of GABA and muscimol (and other agonists such as benzodiazepines and alcohol) on GABAAR and GABABR subunits has been examined by others and has been shown to cause dynamic and complex changes in subunit expression, especially when administered chronically in vivo (Olsen, Neuropharmacology, 2018; Zizzo et al., Neurogastroenterol Motil, 2022; Austrich-Olivares et al., Pharmaceuticals, 2022; Jones et al, Proc Natl Acad Sci, 2022).

Round 2

Reviewer 3 Report

Overall, the authors have done a good job at addressing the concerns raised in my original review. However, there are several minor points that stem from these revisions that need to be corrected.

1. I could not find a mention of Figure 4B in the text. It needs to be discussed.

2. In the Results, the authors state that the effects of GABA on metabolic activity are not significant but in the Discussion (lines 477-478) the authors imply that the results are significant. These different conclusions should be reconciled.

3. Discussion, line 512 doesn't make sense.

Author Response

We thank the reivewer for catching these minor corrections.

  1. Figure 4B has now been mentioned in the Results section.
  2. This sentence has been fixed.
  3. The verb tenses in this sentence have been fixed.